# Differences between Professional and Amateur Cyclists in Endogenous Antioxidant System Profile

**DOI:** 10.3390/antiox10020282

**Published:** 2021-02-12

**Authors:** Francisco Javier Martínez-Noguera, Pedro E. Alcaraz, Raquel Ortolano-Ríos, Stéphane P. Dufour, Cristian Marín-Pagán

**Affiliations:** 1Research Center for High Performance Sport, Campus de los Jerónimos, Catholic University of Murcia, 30107 Murcia, Spain; palcaraz@ucam.edu (P.E.A.); rortolano@ucam.edu (R.O.-R.); cmarin@ucam.edu (C.M.-P.); 2Faculty of Medicine, Translational Medicine Federation (FMTS) UR 3072, University of Strasbourg, 67000 Strasbourg, France; sdufour@unistra.fr; 3Faculty of Sport Sciences, University of Strasbourg, 67084 Strasbourg, France

**Keywords:** catalase, superoxide dismutase, oxidized glutathione, reduced glutathione, hemoglobin, power output

## Abstract

Currently, no studies have examined the differences in endogenous antioxidant enzymes in professional and amateur cyclists and how these can influence sports performance. The aim of this study was to identify differences in endogenous antioxidants enzymes and hemogram between competitive levels of cycling and to see if differences found in these parameters could explain differences in performance. A comparative trial was carried out with 11 professional (PRO) and 15 amateur (AMA) cyclists. All cyclists performed an endogenous antioxidants analysis in the fasted state (visit 1) and an incremental test until exhaustion (visit 2). Higher values in catalase (CAT), oxidized glutathione (GSSG) and GSSG/GSH ratio and lower values in superoxide dismutase (SOD) were found in PRO compared to AMA (*p* < 0.05). Furthermore, an inverse correlation was found between power produced at ventilation thresholds 1 and 2 and GSSG/GSH (r = −0.657 and r = −0.635; *p* < 0.05, respectively) in PRO. Therefore, there is no well-defined endogenous antioxidant enzyme profile between the two competitive levels of cyclists. However, there was a relationship between GSSG/GSH ratio levels and moderate and submaximal exercise performance in the PRO cohort.

## 1. Introduction

Competitive cycling is highly stressful on both aerobic and anaerobic metabolisms. Road cycling races require the riders to produce high relative power output (W/kg) for short duration (i.e., less than 1 min at the start, during steep climb and at the end of the race) while also sustain efforts that last for several minutes to several hours [1]. Overall, professional cyclists (PRO) perform high training volumes (~32,500 Km) during the competitive season, which include 90–100 race days [2]. On the other hand, amateur competitive cyclists (AMA) can be defined as cyclists that train 3–7 times per week, with daily training volumes of 60–120 min and that compete about 20 times in a year [3]. During training sessions and competitions (aerobic and anaerobic exercise), there is a rise in reactive oxygen species (ROS) and subsequent oxidative stress, which can lead to a favorable adaptation in the body’s antioxidant defense system [4]. This improvement in the endogenous antioxidant system is generally associated with lower levels of oxidative stress biomarkers [5].

Within the endogenous antioxidant system, superoxide dismutase (SOD) is the first line of enzymatic defense that transforms the superoxide radical(O_2_^•−^) into hydrogen peroxide (H_2_O_2_) [6]. Then, H_2_O_2_, which is also harmful to cells, can be metabolized in a couple of ways: (1) conversion into water by glutathione peroxidase (GPx) with the reduced glutathione consumption (GSH) being converted into oxidized glutathione (GSSG) and (2) when the production of H_2_O_2_ exceeds the capacity of glutathione peroxidase, catalase (CAT) takes over to remove the excess H_2_O_2_ [7]. However, un-neutralized H_2_O_2_ can interact with transition metals, such as Fe^2+^ and Cu^+^, and result in the production of the hydroxyl radical (^•^OH; i.e., Fenton reaction), which is an extremely powerful oxidizing agent that reacts with all biological macromolecules. However, currently it is not clear what the end product of the Fenton reaction ^•^OH or FeO^2+^ is [8]. Finally, ^•^OH can elicit damage to DNA, oxidation of the thiol group of proteins and peroxidation of lipids [9]. 

Exercise increases oxygen uptake and almost 0.15% of the oxygen consumed can be converted into ROS, which can be harmful to muscle and mitochondrial function [10]. Recent findings show that the main source of ROS during exercise is nicotinamide adenine dinucleotide phosphate oxidases [11]. It is known that long-duration strenuous exercise and extensive sprint training can exceed our ability to detoxify the action of reactive oxygen species within the blood cells, as well as at the muscle level [12]. Conversely, prevention of oxidative stress to enhance performance in professional athletes can be done by the adaptation mechanisms (hormesis) and detoxifying function of antioxidant enzymes (SOD, CAT, GPx, glutathione reductase (GR), glutathione-s-transferase), as well as via non-enzymatic antioxidants (such as vitamins E, A, C, and GSH and GSSG [13]. Therefore, it has been suggested that higher levels of the endogenous antioxidant system may improve performance of skeletal muscle contraction [5]. Cordova et al. [14] analyzed antioxidant markers and showed an average GSH of 3.24 μmol·g^−1^ Hb, GSSG of 1.54 μmol·g^−1^ Hb, GSSG/GSH ratio of 0.56 %, catalase (CAT) of 172.0 mmol·min^−1^·g^−1^ Hb and superoxide dismutase (SOD) of 1983.0 U·g^−1^ Hb activity. In a previous study [15], average values of GSH of ~4.7 μmol·g^−1^ Hb, GSSG of ~0.7 μmol·g^−1^ Hb and GSSG/GSH ratio of 0.15% were observed in PRO cyclists in February in of the same competitive season, suggesting that training season can modify the levels of these components of the endogenous antioxidant system [5,12,15].

At rest, endogenous antioxidant enzymes (EAE) levels are generally lower in athletes than in sedentary subjects, although higher or unchanged levels have been observed [6,16]. Several factors may explain this discrepancy, the most important being differences in the methods used to estimate the state of oxidative stress, the characteristics of the sample population (high-level athlete, sedentary, etc.) and the time of measurement (period of the sport’s season) [13]. However, what is clearly evident is that acute exercise can lead to an imbalance between ROS and endogenous antioxidants, causing what is known as oxidative stress [7]. A recent study has shown that acute exercise at low, moderate or high intensity has the capacity to reduce GSH concentration and increase SOD and CAT activity compared to baseline, in addition to increasing F_2_-isoprostanes (markers of oxidative stress) at all levels of exercise. [17]. In addition, it is generally known that chronic exercise causes an increase in enzymatic and non-enzymatic antioxidant defense, leading to adaptations to the training response and improving the protection against ROS [18]. Chronic exercise (6-week cycling training) has the ability to increase the concentration and activity of GSH, SOD and CAT, while maintaining the levels of F_2_-isoprostanes [17]. This same study also found that moderate and high intensity exercise promoted greater adaptations in antioxidant markers than low intensity exercise at baseline.

Despite the large amount of information on the EAE status and their relationship to the effects of acute and chronic exercise, to our knowledge, there are no studies that have compared the status of antioxidant enzymes between professional and amateur cyclists and their relationship to performance. Therefore, the main objective of this research was to determine the differences in endogenous antioxidants enzymes and hemogram levels between PRO and AMA, and whether these might be related to differences in performance (power output at VT1, VT2 and VO_2MAX_) in an incremental test. Finally, the secondary objective was to assess whether differences between endogenous antioxidant enzymes and hematological were associated with differences in performance between PRO and AMA.

## 2. Methodology

### 2.1. Selection of Participants

A total of 26 male cyclists (11 PRO, 15 AMA) were recruited and completed the study. The PRO were competing at the *Union Cycliste Internationale* (UCI) PRO TOUR level and have participated in UCI major stage races (*Vuelta a España, Giro d’Italia, Tour de France*). The 15 AMA were from the southeast region of Spain. The PRO riders were selected based on the following criteria: (1) 20 to 40 years of age, (2) enrolled in a professional licensed team and (3) competed in at least one of the main 3-week stage races in the last years. Subjects for the AMA group had to meet the following inclusion criteria: (1) 20 to 40 years of age, (2) had at least 3 years of cycling experience and (3) performed specific training 6–12 h/week. 

All subjects signed the informed consent document before their participation. The study was performed following the guidelines of the Helsinki Declaration for Human Research [19] and was approved by the Ethics Committee of the Catholic University of Murcia (CE091802).

### 2.2. Study Protocol

The experimental design of the study required each rider to visit the laboratory twice between the end of October and December (i.e., post-season period). In the first visit, a medical exam and blood analysis were completed to check their state of health. In the second visit (post-48 h), the cyclists performed a maximal incremental test. The 2 h prior to this latter test, they ingested a standardized breakfast, which was based relative to body mass (557.7 kcal) and composed of 95.2 g of carbohydrates (68%), 19.0 g of protein (14%) and 11.3 g of lipids (18%), established by a sports nutritionist. All subjects were instructed to refrain from high-intensity training 48 h before each visit.

### 2.3. Incremental Test

An incremental step test with final ramp until exhaustion was performed on a cycle ergometer (Cyclus 2TM, RBM elektronik- automation GmbH, Germany) using a metabolic cart (Metalyzer 3B. Leipzig, Germany) to determine VT1, VT2 and VO_2max_, as well as the associated levels of power output. The testing protocol started with 35 W and increased by 35 W every 2 min until RER > 1.05 was reached, from which the final ramp (+35 W·min^−1^) until exhaustion was initiated [20]. To ensure that VO_2MAX_ was achieved, at least 2 of the following criteria had to be met: plateau in the final VO_2_ values (increase ≤ 2.0 mL·kg^−1^·min^−1^ in the two last loads), maximal theoretical HR (220-age)·0.95) [21], RER ≥ 1.15 and lactate ≥ 8.0 mmol·L^−1^ [22,23]. Ventilatory thresholds were obtained using the ventilatory equivalents method described by Wasserman [24]. 

### 2.4. Blood Analysis

A total of 21.5 mL of blood were withdrawn from the antecubital vein for analyses: one 3.0 mL tube with ethylenediaminetetraacetic acid (EDTA) for hemogram and another 3.5 mL tube with polyethylene terephthalate (PET) for biochemical parameters. For the measurement of antioxidant parameters, five 3.0 mL EDTA tubes were obtained, where one tube was immediately centrifuged at 3500 rpm at 4 °C for 10 min. All tubes were temporarily stored at 2–4 °C and then sent to an external laboratory for analysis. Red blood cell count was carried out in an automated Cell-Dyn 3700 analyser (Abbott Diagnostics, Chicago, IL, USA) using internal (Cell-Dyn 22) and external (Program of Excellence for Medical Laboratories-PEML) controls. Values of erythrocytes, hemoglobin, hematocrit and hematometra indexes (mean cell volume (MCV), mean cell hemoglobin (MCH) and mean cell hemoglobin concentration (MCHC)) were estimated.

### 2.5. Oxidative Stress and Antioxidant Status Markers

#### 2.5.1. Catalase

The activity of catalase was measured in the whole blood using a UV-VIS spectrophotometer. The catalase enzyme extracts the peroxides from the region of the gel it occupies, following the isolation of the native protein. The removal of peroxide does not cause potassium ferricyanide (yellow substance) to be reduced to potassium ferrocyanide, which reacts with ferric chloride to form a blue Prussian precipitate. The catalase positive control activity is defined in international unit equals (1 unit) to the amount of catalase necessary to decompose 1.0 μM of H_2_O_2_ per minute at pH 7.0 at 25 °C while H_2_O_2_ concentration falls from ≈ 10.3 mM to 9.2 mM. The absorbance of H_2_O_2_ decreases at 240 nm proportional to its decomposition so that the concentration of H_2_O_2_ is critical in this determination. The decrease in absorbance per time unit is the measure of catalase activity [25]. Results were expressed in U/g of Hb.

#### 2.5.2. SOD

Superoxide dismutase (SOD) activity was measured using an SD125 Ransod kit (Randox Ltd. Crumlin, Reino Unido) in whole blood. Xanthine and xanthine oxidase were used to produce superoxide anion (O_2_^•−^), which responded with the 2-(4-iodophenyl)-3-(4-nitrophenol)-5-phenyltetrazolium chloride (INT) reactive and formed a red complex that is detectable at 420 nm. The SOD activity was measured as the inhibition degree of this reaction [26]. Results were expressed in U/g of Hb.

#### 2.5.3. Glutathione

The analysis of reduced glutathione (GSH) was performed using the glutathione-S-transferase assay described by Akerboom and Sies [27]. Calculation of GSH was performed from lymphocytes treated with perchloric acid at a final concentration of 6%, collecting the supernatant after vortexing and subsequent centrifugation for 10 min at 10,000 rpm. Following collection of the supernatants in vials, high-performance liquid chromatography (HPLC) coupled to a Waters NH2 ODS S5 column (0.052, 25 cm) was conducted. Oxidized glutathione (GSSG) was analyzed using a similar method described by Asensi. [28].

### 2.6. Statistical Analyses

The statistical analysis was performed using the Statistical Package for Social Sciences (SPSS 21.0, International Business Machines Chicago, IL, USA). Descriptive statistics are presented as mean ± standard deviation (SD). Levene and Shapiro–Wilks tests were performed to check for homogeneity and normality of the data, respectively. A Student’s t-test for unpaired data was used to evaluate differences between groups. Additionally, the standardized mean differences were calculated using Cohen’s effect size (ES) (95% confidence interval) for all comparisons. Threshold values for ES statistics were as follows: >0.2 small, >0.5 moderate, >0.8 large [29]. The different correlations between the parameters were evaluated using Pearson’s correlation (r). Significance level was set at *p* ≤ 0.05.

## 3. Results

### 3.1. Subject Characteristics

The general characteristics and hemogram results are presented in Table 1. Age, body mass and height were not different between PRO and AMA groups. Interestingly, PRO had higher MCH (4.8%, *p* < 0.001) and MCHC (3.6%, *p* < 0.001) compared to AMA. There were no group differences in RBC, Hb, HCT and MCV (Table 1). No correlation was found between age and antioxidant markers in both groups. However, there were correlations found between age and Hb, HCT and MCHC (r ≤ −0.597, *p* < 0.05).

### 3.2. Antioxidant Parameters 

Table 2 shows the outcomes of CAT, SOD, GSSG, GSH, %GSSG/GSH and GSSG+GSH, which were measured at baseline before the incremental tests. Higher levels in CAT (30.0%, *p* < 0.001), GSSG (63.2%, *p* < 0.001) and %GSSG/GSH (70.1%, *p* < 0.001), and lower levels in SOD (−16.2%, *p* = 0.009) were found in PRO compared to AMA. However, no differences in GSH (−4.3%, *p* = 0.216) and GSSG+GSH (−3.5%, *p* = 0.317) values were observed between PRO and AMA.

### 3.3. Physiological and Metabolic Parameters at VT1

VO_2_, W, WR, %VO_2MAX_, HR and RER values at VT1 are shown in Table 3. Significant group differences in VO_2_ (76.0%, *p* < 0.001), W (90.4%, *p* < 0.001), WR (92.5%, *p* < 0.001), %VO_2MAX_ (53.3%, *p* < 0.001) and HR (12.9%, *p* = 0.004), but not for RER (0.78%, *p* = 0.707) were observed.

GSSG/GSH was significantly correlated with W_VT1_ and VO_2VT1_(r = −0.657 and r = −0.651; *p* < 0.05, respectively) in PRO (Table 4) (Figure 1).

### 3.4. Physiological and Metabolic Parameters at VT2

Table 3 demonstrates the VT2 results of VO_2_, W, WR, %VO_2MAX_, HR and RER. Significant group differences in VO_2VT2_ (25.6%, *p* = <0.001), W_VT2_ (32.5%, *p* = <0.001), WR_VT2_ (34.1%, *p* < 0.001) and %VO_2MAXVT2_ (6.6%, *p* = 0.005) were observed.

GSSG/GSH was significantly correlated with W_VT2_ and VO_2VT2_(r = −0.635 and r = −0.622; *p* < 0.05, respectively) in PRO (Figure 1). GSSG tended to correlate with W_VT2_ (r = −0.575; *p* = 0.06) in PRO (Table 4).

### 3.5. Physiological and Metabolic Parameters at VO_2max_

Maximal values of VO_2_, VO_2_/R, W, WR, HR and RER are presented in Table 3. Significant group differences in VO_2MAX_ (15.9%, *p* = 0.002), VO_2_/R_MAX_ (17.5%, *p* = 0.002), W_MAX_ (23.8%, *p* < 0.001), WR_MAX_ (25.8%, *p* < 0.001), and RER_MAX_ (7.0%, *p* = 0.001), but not for HR_MAX_ (*p* = 0.966) were found. 

In VO_2MAX_, no correlation with any antioxidant marker was observed (Table 4).

## 4. Discussion

This study provides the first direct comparison of endogenous antioxidant, hematological, performance and metabolic biomarkers (VT1, VT2 and VO_2MAX_) between PRO and AMA cyclists. Our results demonstrate that: (i) PRO have higher values in MCH, MCHC, CAT, GSSG and GSSG/GSH but lower values in SOD than AMA; (ii) PRO have higher levels of absolute and relative power output and oxygen consumption in all intensity zones (VT1, VT2 and VO_2MAX_) than in AMA, with the largest differences found at VT1; (iii) inverse correlations were identified in W_VT1_, VO_2VT1_, W_VT2_ and VO_2VT2_ with GSSG/GSH in PRO.

### Differences in Antioxidant Enzymes and Hemogram

When intense physical exercise is performed (especially in untrained or those not familiar with the exercise), there is an increase in the production of reactive oxygen species, which are neutralized by our complex endogenous antioxidant defense system (GSH, GSSG, CAT, SOD, GPx and GR) and by exogenous antioxidants (vitamin C, vitamin E, carotenes) [30]. 

Regarding EAE, we observed higher levels of CAT activity, GSSG and GSSG/GSH, but lower levels of SOD activity in PRO versus AMA. Mena et al. [31] found higher resting levels of SOD, CAT and GPx in a sample of PRO cyclists compared to sedentary people. Tauler et al. [32] also showed differences in antioxidant enzyme activity in erythrocyte between PRO and AMA at rest. In the same study, a decrease in CAT (−12%), GPx (−14%) and GR activity (−16%) but an increase in SOD activity of about 25% after a submaximal test (80% VO_2MAX_; 1 h 30 min) was reported [32]. 

Long distance runners have been shown to have a three-fold higher CAT activity compared to short distance runners [18]. Similarly, it was observed that marathon runners had twice as high catalase activity compared to sprinters [12]. In this study, we also demonstrated higher levels of CAT in PRO than in AMA, and this may be largely explained by the fact that PRO perform greater volume, intensity and competitions (higher aerobic load and prolonged periods of exercise) than AMA, which induces higher levels of exposure to ROS and, consequently, adaptations of EAE [6]. When CAT levels increase, it is possible that GPx activity is not sufficient to neutralize high levels H_2_O_2_ (endurance exercise) [7].

Regarding SOD, Mena et al. [31] observed lower levels of SOD activity (−32.1%) in PRO than in elite cyclist, but in the case of CAT (80.0%) and GPx (149.0%), the levels were higher, reporting an ascending behavior of SOD during a stage race (2800 km in 17 stages) in PRO. Tauler et al. [32] has also found lower levels of SOD activity in PRO (−19.8%) than in AMA at baseline, which are in line with the results of our study. Antioxidant enzyme activity can be modified either by an initial increase (adaptation) or a decrease if the oxidative stress of long duration (utilization) [33]. Therefore, the low basal levels of SOD activity in professional cyclists could be overwhelmed and the high concentration of superoxide anions could activate CAT, allowing compensated metabolization of H_2_O_2_. This may be the reason why PRO has lower levels of SOD activity than AMA, as PRO have higher levels of exercise exigency that sometimes get close to exhaustion, which can lead a decrease in the working capacity of SOD.

On the other hand, there is evidence to suggest that GSH or GSH/GSSG decreases during exercise because of its utilization against ROS [33]. Ultra-endurance exercise depletes erythrocyte GSH levels by ∼66% for 24 h and levels remain ∼33% lower than normal 1 month later [34]. PRO frequently compete in longer distance events than AMA, which can lead to lower levels of GSH in PRO than in AMA, although no differences were observed in GSH between PRO vs. AMA in our study. In addition, the muscle can import GSH from plasma during exercise, and as a result, there is a change in the GSH/GSSG ratio after exercise with a decrease in the GSH/GSSG ratio at the time of exhaustion [35]. Furthermore, it is important to mention that tissues are not only capable of importing GSH but also exporting GSSG under oxidative stress [35]. Moreover, GSH is a molecule that is key in cellular redox status regulation, and consequences of prolonged GSH depletion may include a compromise in immunity, where lower GSH is associated with decreased lymphocyte proliferation and increased viral reactivation [34]. 

GSSG levels are a biomarker of cellular oxidative stress, since GSH is an important antioxidant in many tissues and oxidizes in the catalyzed reduction of H_2_O_2_ to H_2_O to become GSSG [36]. The increase in GSH (mainly) and GSSG in plasma after exercise could be explained by an efflux from the liver to other tissues, including skeletal muscle [37]. GSSG levels in skeletal muscle have previously been shown to increase by ~50% in rats after running on a treadmill at moderate intensity [38] and by ~20% after cycling in humans (workload corresponding to 90% of VO_2_peak; 10 × 4 min) [39]. Leonardo et al. [40] observed an increase in both GSSG and GSSG/GSH after a period of intense PRO training, which returned to their baseline levels after a period of tapering. We found similar baseline values of GSSG in our study. In addition, we found higher levels of GSSG and GSSG/GSH in PRO than in AMA. 

The efforts made during cycling competitions produce oxidative stress in lymphocytes, leading to a reduction in GSH levels and an increase in GSSG levels. The decrease in GSH and increase in GSSG during exercise may be explained by an increase in H_2_O_2_ formation, as reported by Wang et al. who found that high-intensity exercise (80% VO_2MAX_) decreased GSH levels while lipid peroxidation increased immediately and after 24h of exercise [41]. Furthermore, in this study, lymphocytes were incubated with H_2_O_2_ for 2 and 4 h, promoting an increase in DNA fragmentation immediately and 24 h after high intensity exercise. Thus, H_2_O_2_ would cause a failure of the endogenous antioxidant system leading to DNA damage in lymphocytes. Ferrer et al. [42] found that high intensity exercise (swimming) increased GPx activity (converts GSH to GSSG) in lymphocytes, in the same way as other authors found after a cycling stage [43,44]. This supports the decrease in GSH and increase in GSSG after high intensity exercise. Therefore, the higher levels of GSSG and GSSG/GSH in PRO vs. AMA in our study may be due to a higher production of ROS, which leads to a higher production of GSSG and, consequently, of GSSG/GSH together with a decrease in GSH.

In addition, our study is the first to show correlations between GSSG/GSH with W_VT1_ (r = −0.657) and W_VT2_ (r = −0.635) in PRO. This is also supported by a trend towards a significant correlation between GSSG and WVT2 (r = −0.575; *p* = 0.06) in PRO. These relationships suggest that cyclists who generate more power at VT1 and VT2 have lower GSSG/GSH levels, and therefore, less oxidative stress, as GSSG/GSH ratio is known to be a marker of antioxidant status [20].

In response to strenuous physical working conditions, the body’s antioxidant capacity may be temporarily diminished, as its components are used to scavenge the harmful radicals that are produced [45]. It is well known that exercise-induced ROS are detrimental to physiological function, including decreased performance and immune function and increased fatigue [45]. Moreover, it has been shown that the response of antioxidant capacity to exercise responds in a similar way to the activity of EAE [45]. Therefore, the antioxidant defense system may be temporarily reduced in response to increased ROS production but may increase during the recovery period as a result of the initial prooxidant insult [46]. However, contradictory findings have been reported where increases in GPx, SOD, and CAT, as well as decreases in GPx, GR, SOD have been observed [45]. Evidently, this controversy may depend on the moment of sampling (i.e., period of the season), as well as on the duration and intensity of the exercise, which varies considerably between studies. 

It could be that there is an undefined optimal level of ROS production and oxidative damage required for adaptations in antioxidant defenses and other physiological parameters, leading to health and performance improvements [45]. However, overproduction of ROS and oxidative damage due to chronic long-term exercise and/or overtraining may exceed the above-mentioned optimal level, resulting in irreparable oxidative damage, which can lead to the development or progression of poor health and/or disease [47]. Therefore, the measurement of the antioxidant capacity (CAT, SOD, GSH, GSSG and GSGG/GSH) of the body is used as a marker of oxidative stress and can provide us insight on how it affects performance. Given the results of our research and the evidence shown in the scientific literature, there is no endogenous antioxidant profile defined in PRO compared to AMA. 

There are also other antioxidant proteins, such as peroxiredoxin (PRX) and thioredoxin (TRX) containing thiol groups, with a high capacity to neutralize reactive oxygen and nitrogen species and decrease oxidative stress [48]. One study showed how moderate and high-intensity exercise and a low volume high intensity interval training trial increased TRX (85%, 64% and 206%, respectively); however, PRX only increased during high intensity exercise (moderate: −6229%; high: 203% and low volume high intensity interval: −23%, respectively) in peripheral blood mononuclear cells [48]. In addition, an increase in nuclear transcription factor kappa B was found during all exercises, suggesting an activation of the inflammatory system, probably due to increased oxidative stress. Future studies should examine whether there are differences in these antioxidant proteins between PRO and AMA and their relationship with performance.

Regarding hematological parameters, no significant differences were found except for MCH and MCHC between PRO vs. AMA. Schumacher et al. found hematological values in elite cyclists from the German national team (blood samples collected between November and January) and the values were similar to ours in Hb (~15.5 g/dL), Hct (~45.0%) and RBC (~5.0 × 10^6^/mm^3^) in PRO [49]. In addition, other studies have found hematological values of approximately 15.0 g/dL of Hb and 45% of Hct in professional cyclists [50,51,52]. Well-trained cyclists have found values of 14.3 g/dL in Hb and 43.1% in Hct, values lower than PRO [53]. However, Bejder et al. [54] observed amateur competitive cyclist values of 14.8 g/dL Hb, 42.8% Hct, 4.92 × 10^6^·μL^−1^RBC, 87.1 fl MCV, 30.1 pg MCH and 34.6 g/dL MCHC, lower than those reported in PRO.

MCH indicates the amount of hemoglobin contained in an erythrocyte and MCHC is the average hemoglobin concentration [55]. Therefore, the red blood cells of PROs will have a higher oxygen transport capacity due to the higher levels of MCH and MCHC. Currently, no study on cyclists has examined the differences in MCH and MCHC, so we cannot draw many conclusions in this regard. These hematological parameters have mainly been used as markers of anemia both in athletes and in the general population [56], but so far, they are not associated with an athlete’s performance level in this study. 

## 5. Limitations

Our study had limitations with regards to the sample number, since it was more difficult to recruit PRO athletes than lower-level athletes (AMA). 

Differences in endogenous antioxidant marker between this study and previous works may be influenced by the instrumentation and methodology used, the timing of the season at which the measurements were made, and the training status of the cyclists. 

## 6. Conclusions

Regarding the endogenous antioxidants profile, PRO had higher values of CAT, GSSG and GSSG/GSH compared to AMA. An inverse correlation was found for the first time between W_VT1_ and W_VT2_ with GSSG/GSH at rest only in PRO. This indicates better antioxidant status that allow for higher performance with regard to power output. Future studies should examine how training adaptations affect the studied variables and how antioxidant enzymes evolve during a race stage (e.g. Tour de France), in order to see their association with performance, recovery and fatigue, thereby helping to develop monitoring tools for medical doctors, nutritionists and coaches.

## Figures and Tables

**Figure 1 antioxidants-10-00282-f001:**
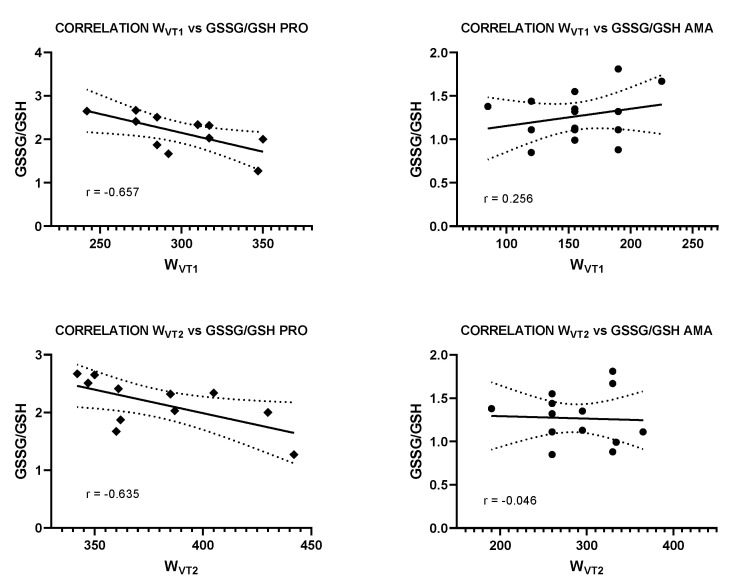
Correlations between the power generated at the ventilation threshold 1 and 2 between the GSSS/GSH ratio in PRO and AMA.

**Table 1 antioxidants-10-00282-t001:** Baseline general characteristics and hemogram variables of professional and amateur cyclists.

	PRO	AMA	*p*-Value	Cohen’s d	Effect Size
Age (years)	28.3 (4.65)	29.3 (6.54)	0.671	0.17	Trivial
Body mass (kg)	68.5 (4.43)	69.9 (5.50)	0.488	0.28	Small
Height (cm)	178.0 (6.93)	175.0 (6.71)	0.274	0.44	Small
HEMOGRAM					
RBC (10^6^·μL^−1^)	5.06 (0.281)	5.15 (0.260)	0.441	0.08	Trivial
Hb (g·dl^−1^)	15.6 (0.827)	15.1 (0.676)	0.107	0.49	Small
HCT (%)	44.5 (2.28)	44.6 (1.57)	0.866	0.13	Trivial
MCV (fl)	87.9 (2.19)	86.8 (2.92)	0.305	1.10	Large
MCH (pg)	30.8 (0.35)	29.4 (1.03)	<0.001	1.44	Large
MCHC (%)	35.0 (0.74)	33.8 (0.60)	<0.001	1.19	Large

Values are expressed as mean (SD). Abbreviations: RBC = red blood cell; Hb = hemoglobin; HCT = hematocrit; MCV = mean corpuscular volume; MCH = mean corpuscular hemoglobin; MCHC = mean corpuscular hemoglobin concentration and SD = standard deviation.

**Table 2 antioxidants-10-00282-t002:** Endogenous antioxidant enzymes from professional and amateur cyclists.

	PRO	AMA	*p*-Value	Cohen’s d	Effect Size
CAT (U/g Hb)	32.5 (5.34)	25.0 (4.51)	<0.001	1.55	Large
SOD (U/g Hb)	1213 (233.0)	1447 (184.4)	0.009	1.13	Large
GSSG (nmol/mg protein)	0.524 (0.103)	0.321 (0.077)	<0.001	2.28	Large
GSH (nmol/mg protein)	24.4 (2.00)	25.5 (2.17)	0.216	0.50	Moderate
GSSG/GSH	2.16 (0.436)	1.27 (0.279)	<0.001	2.52	Large
GSSG+GSH (nmol/mg protein)	24.9 (2.02)	25.8 (2.19)	0.317	0.41	Small

Values are expressed as mean (SD). Abbreviations: CAT = catalase; SOD = superoxide dismutase; GSH = reduced glutathione; GSSG = oxidized glutathione; % GSSG/GSH = oxidized/reduced glutathione ratio and SD = standard deviation.

**Table 3 antioxidants-10-00282-t003:** Metabolic and performance variables of professional and amateur cyclists.

	**PRO**	**AMA**	***p*** **-Value**	**Cohen’s d**	**Effect Size**
**VT1**					
VO_2_ (mL·min^−1^)	3593 (271.0)	2041 (401.0)	<0.001	4.40	Large
W	299 (32.9)	157 (36.1)	<0.001	4.07	Large
WR (W·kg^−1^)	4.37 (0.42)	2.27 (0.56)	<0.001	4.14	Large
%VO_2max_	76.2 (3.91)	49.7 (5.58)	<0.001	5.36	Large
HR (beats·min^−1^)	149 (14.7)	132 (13.2)	0.004	1.25	Large
RER	0.906 (0.05)	0.899 (0.04)	0.707	0.15	Trivial
**VT2**					
VO_2_ (mL·min^−1^)	4259 (234.0)	3389 (505.0)	<0.001	2.10	Large
W	379 (34.0)	286 (45.1)	<0.001	2.28	Large
WR (W·kg^−1^)	5.54 (0.41)	4.13 (0.74)	<0.001	2.28	Large
%VO_2max_	90.3 (2.36)	84.7 (5.67)	0.005	1.24	Large
HR (beats·min^−1^)	168 (11.1)	171 (9.4)	0.467	0.29	Small
RER	1.01 (0.05)	1.03 (0.03)	0.323	0.40	Small
**VO_2max_**					
VO_2_ (mL·min^−1^)	4714 (241.0)	4066 (580.7)	0.002	1.38	Large
VO_2_/R (mL·kg^−1^·min^−1^)	69.0 (3.94)	58.7 (9.58)	0.003	1.34	Large
W	474 (31.5)	383 (49.2)	<0.001	2.13	Large
WR (W·kg^−1^)	6.93 (0.44)	5.51 (0.81)	<0.001	2.09	Large
HR (beats·min^−1^)	186 (7.42)	186 (7.62)	0.966	0.02	Trivial
RER	1.22 (0.04)	1.14 (0.06)	0.001	1.49	Large

Values are expressed as mean (SD). Abbreviations: VO_2_ = oxygen uptake; VO_2max_ = maximum oxygen consumption; VO_2_/R = maximum oxygen consumption relative to weight; W = power output; WR = power output relative to weight; %VO_2max_ = percentage of VO_2max_; HR = heart rate (beats·min^−1^); RER = respiratory exchange ratio; VT1 = ventilatory threshold 1; VT2 = ventilatory threshold 2 and SD = standard deviation.

**Table 4 antioxidants-10-00282-t004:** Correlation between endogenous antioxidant enzymes and performance-metabolic variables from professional and amateur cyclists.

		**CAT**	**SOD**	**GSSG**	**GSH**	**%GSSG/GSH**	**GSSG + GSH**
**PRO (*n* = 11)**							
W_VT1_	r	−0.120	0.305	−0.449	0.425	**−0.657**	0.397
	*p*-value	0.72	0.36	0.17	0.19	**0.03**	0.23
VO_2VT1_	r	0.001	0.378	−0.442	0.457	**−0.651**	0.429
	*p*-value	0.998	0.252	0.173	0.157	**0.030**	0.188
W_VT2_	r	−0.253	0.183	−0.575	0.116	**−0.635**	0.085
	*p*-value	0.45	0.59	0.06	0.73	**0.04**	0.80
VO_2VT2_	r	−0.319	0.423	−0.518	0.277	−0.622	0.247
	*p*-value	0.34	0.20	0.10	0.41	0.04	0.46
W_MAX_	r	−0.045	0.186	−0.342	0.239	−0.443	0.219
	*p*-value	0.90	0.58	0.30	0.48	0.17	0.52
VO_2MAX_	r	−0.375	0.422	−0.312	0.304	−0.414	0.284
	*p*-value	0.26	0.20	0.35	0.36	0.21	0.40
**AMA (*n* = 15)**							
W_VT1_	r	0.181	0.172	0.206	−0.102	0.256	−0.098
	*p*-value	0.52	0.54	0.46	0.72	0.36	0.73
VO_2VT1_	r	0.360	0.159	0.182	−0.108	0.230	−0.105
	*p*-value	0.19	0.57	0.52	0.70	0.41	0.71
W_VT2_	r	0.414	0.113	−0.002	0.047	−0.046	0.040
	*p*-value	0.13	0.69	0.99	0.87	0.87	0.89
VO_2VT2_	r	0.358	0.234	−0.104	−0.068	−0.097	−0.077
	*p*-value	0.19	0.69	0.71	0.81	0.73	0.78
W_MAX_	r	0.180	0.173	−0.136	−0.379	0.009	−0.386
	*p*-value	0.52	0.54	0.63	0.16	0.97	0.16
VO_2MAX_	r	0.289	0.278	−0.118	−0.334	0.001	−0.339
	*p*-value	0.30	0.32	0.66	0.22	0.10	0.22

Values are expressed as mean (SD). Abbreviations: CAT = catalase (U/g Hb); SOD = superoxide dismutase (U/g Hb); GSH = reduced glutathione (nmol/mg protein); GSSG = oxidized glutathione (nmol/mg protein); % GSSG/GSH = oxidized/reduced glutathione ratio; VO_2_ = oxygen uptake; VO_2MAX_ = maximum oxygen consumption; VT1 = ventilatory threshold 1; VT2 = ventilatory threshold 2 and W = power output

## Data Availability

All data is contained within the article.

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
