# Peer review of "Differences between Professional and Amateur Cyclists in Endogenous Antioxidant System Profile"

_antioxidants, 2021, doi:10.3390/antiox10020282_

Round 1

Reviewer 1 Report

-Better explain the reasons for the choice of this  specific antioxidants panel between the many possible. The inclusion of biomarkers of the oxidant counterpart would have enriched the study. also some inflammatory parameters related to inflammation (some easily calculated by hemocrome) wolud be easily calculated and evaluated in their relationship with the other antioxidant, physiologic and metabolic parameters. If possible add these parameters, or at least discuss these issues

the results are well presented, the topic is of interest to readers of Antioxidants. However, the novelty of the results is not high. The choice of the antioxidants evaluated (why not others?) Must be justified. If it is possible to add other biomarkers of the oxidizing counterpart, and perhaps some simple biomarkers of inflammation (also from the blood count), which can help to better understand the complex relationship of the redox system

Author Response

Reviewer 1

We thank the reviewer for their constructive and helpful feedback on our manuscript. We have replied to each specific comment in the section below and have introduced the corresponding edits into the manuscript using Word’s track changes.

Better explain the reasons for the choice of this specific antioxidants panel between the many possible. The inclusion of biomarkers of the oxidant counterpart would have enriched the study. Also, some inflammatory parameters related to inflammation (some easily calculated by hemocrome) wolud be easily calculated and evaluated in their relationship with the other antioxidant, physiologic and metabolic parameters. If possible add these parameters, or at least discuss these issues.

Response: We appreciate the suggestion to incorporate other markers, both antioxidant and inflammatory (hemocrome), which we will consider for future research. However, the antioxidant markers listed in the manuscript were chosen to be able to compare our results with other studies. We also had a limited budget, which prevented us to incorporate other antioxidant parameters, such as glutathione peroxidase and reductase, thioredoxin, peroxiredoxins, and non-enzymes, including vitamins C and E, retinol, bilirubin, uric acid, redox glutathione, thiols, coenzyme Q10, stress proteins, albumins, and pro-oxidant activity with protein carbonyl, malondialdehyde, isoprostanes, and 8-OHdG.

Response: Following your suggestion, we have included other antioxidant parameters not analysed in our sample in the discussion. Line 380-390.

The results are well presented, the topic is of interest to readers of Antioxidants. However, the novelty of the results is not high. The choice of the antioxidants evaluated (why not others?) Must be justified. If it is possible to add other biomarkers of the oxidizing counterpart, and perhaps some simple biomarkers of inflammation (also from the blood count), which can help to better understand the complex relationship of the redox system

Response: Please see our answer above regarding the choice of antioxidant markers. In addition, other parameters not analyzed in our sample have been included in the discussion, as suggested.

Author comment: We appreciate all the comments made on our manuscript, which helped improve it’s quality.

Reviewer 2 Report

In this paper, authors analyse differences in endogenous antioxidants enzymes between professional and amateur cyclists. I found this manuscript very well written and all data are well presented.

I've got only a question for authors:

Study protocol: I could be interesting to have information about their regular diet in both PRO and AMA. Can authors give this information??? At professional level, athletes tend to follow particular and more specific diets. Moreover, it should be stated if subjects utilized food supplements based on antioxidants.

Author Response

Reviewer 2

In this paper, authors analyse differences in endogenous antioxidants enzymes between professional and amateur cyclists. I found this manuscript very well written and all data are well presented.

Response:  We thank the reviewer for their constructive and helpful feedback on our manuscript. We have replied to each specific comment in the section below and have introduced the corresponding edits into the manuscript using Word’s track changes.

I've got only a question for authors:

Study protocol: I could be interesting to have information about their regular diet in both PRO and AMA. Can authors give this information??? At professional level, athletes tend to follow particular and more specific diets. Moreover, it should be stated if subjects utilized food supplements based on antioxidants.

Response:  We have information about the regular diet of AMA but the PRO, as this latter group was collected by the team’s nutritionist and, in the end, there were problems between the team and the nutritionist for which we can no longer obtain this information. Our intention was to include this data. None of the cyclists were taking any medication or supplements, as we are aware that antioxidant supplements (supplements in general) can influence the levels of antioxidant enzymes or molecules with antioxidant activity.

Author comment: We appreciate all the comments made on our manuscript, which helped improve it’s quality.

Reviewer 3 Report

Dear Editors,

The article presented to me for review is generally well prepared and may be of interest to Antioxidants readers as it concerns the research of an interesting group. Unfortunately, this is not a large group (as the authors emphasize in their criticism of the method), therefore it is very important that the characteristics of the groups do not raise any doubts. I have reservations about the information in the article about matching study participants to two groups. My question was whether the groups differed in comorbidities, BMI data would be useful if both weight and height are given. Maybe the concentration of glucose and creatinine were assessed - to see if there were any significant differences in this range ... etc?

There is generally not enough information as to whether the people included in the study were healthy.

Some chronic diseases do not rule out being an athlete at all. The lack of such data does not allow to fully refer to the results obtained by the authors. Besides, I have no critical comments. The study was well planned and conducted. The results were presented in a comprehensible and transparent manner, apart from the characteristics of the group that I have already mentioned. Conclusions resulting from the obtained results The discussion is interesting and based on the results.

After clarifying the health status of the study participants, I believe that the publication will be suitable for printing in such a good journal as Antioxidants.

Best regards

Dorota Formanowicz

Author Response

Reviewer 3

The article presented to me for review is generally well prepared and may be of interest to Antioxidants readers as it concerns the research of an interesting group. Unfortunately, this is not a large group (as the authors emphasize in their criticism of the method), therefore it is very important that the characteristics of the groups do not raise any doubts. I have reservations about the information in the article about matching study participants to two groups. My question was whether the groups differed in comorbidities, BMI data would be useful if both weight and height are given. Maybe the concentration of glucose and creatinine were assessed - to see if there were any significant differences in this range ... etc?

Comment:  We thank the reviewer for their constructive and helpful feedback on our manuscript. We have replied to each specific comment in the section below and have introduced the corresponding edits into the manuscript using Word’s track changes.

Response: We did not incorporate glucose and creatinine because we had a very small budget to perform this study and had to select the antioxidant biochemical parameters that appear in the manuscript. Our idea was to incorporate other biochemical parameters to give more power to the study, but antioxidant markers are very expensive. Regarding the sample, we would have liked to include more subjects, but it is very difficult to recruit professional cyclists, in addition to having a limited budget. For future studies, we will follow your suggestions and include glucose and creatinine.

There is generally not enough information as to whether the people included in the study were healthy.

Response: All subjects were healthy, as discussed in lines 126. We have included that a medical examination and blood draw were performed to ensure the cyclist’s health status.

Some chronic diseases do not rule out being an athlete at all. The lack of such data does not allow to fully refer to the results obtained by the authors. Besides, I have no critical comments. The study was well planned and conducted. The results were presented in a comprehensible and transparent manner, apart from the characteristics of the group that I have already mentioned. Conclusions resulting from the obtained results The discussion is interesting and based on the results.

Response: As mentioned above, the subjects did suffer from any pathology of any kind. We took into account that certain pathologies or habits, such as smoking, can modify the endogenous antioxidant profile in humans.

 Author comment: We appreciate all the comments made on our manuscript, which helped improve it’s quality.

Reviewer 4 Report

The submitted work aimed to investigate the potential differences between professional and amateur cyclist in terms of endogenous antioxidant enzymes profile. Based on their findings, the authors argue that there is no well-defined endogenous antioxidant enzyme profile between the two levels of cyclists, as demonstrated by the higher CAT, GSSG and GSSG/GSH ratio and the lower SOD levels between PRO compared to AMA. With regard to performance, within the professional group, there was a reverse relationship between GSSG/GSH ratio levels and moderate and submaximal exercise performance. Please find next some comments on the paper:

Title: (1) I think that an “s” is needed in “cyclist”; (2) the title refers to antioxidant enzymes profile, however, two enzymatic (CAT and SOD) and two non-enzymatic (GSH and GSSG as well as their ratio) parameters were measured. Probably, the title should be revised accordingly.

Abstract: (1) what does the word “parallel” stand for? (2) use the abbreviations PRO and AMA in their first mention.

General comment: the authors argue that a main aim of the study was to investigate “…if differences found in these parameters [antioxidant enzymes] could explain differences in performance”. Taking into account that only some correlation analyses were performed and that the theoretical background for such a cause-and-effect relationship is lacking in the introduction, I think that it is vague to argue that a real relationship can be established in the context of the present study. Certainly, differences in performance between these two groups can definitely be explained by other factors apart from redox state.

There are several issues in the introduction as regards to the information about (exercise-induced) redox processes:

(1) The references provided are sometimes either not suitable or outdated. For the health- and redox-related benefits of chronic exercise check PubMedIDs: 32174127 and 32192916.

(2) The increased oxygen uptake during exercise is currently regarded not the main reason for ROS production during exercise (and thus not mitochondria, but NADPH oxidases) (PubMedID: 23915064).

(3) 2% of the oxygen consumed is not converted to ROS. The real percentage is much lower, approximately equal to 0,13% to 0,15% (PubMedID: 12237311).

(4) Based on kinetic rates, peroxiredoxins are currently regarded the most efficient antioxidant enzymes against hydrogen peroxide and they also serve for signaling purposes (PubMedID: 20919930). This family of antioxidant enzymes are neglected in the present study.

(5) Reactive species are in the present manuscript widely regarded as detrimental molecules. Even hydrogen peroxide is mentioned as “…also harmful to cells…”. However, we now know that this is probably the most well-known reactive species, along with nitric oxide, featuring signaling properties (PubMedID: 24634836).

(6) As the authors correctly mention, Fenton reaction requires free iron. However, this is probably not the case during exercise and in health in general, taking into account that biological systems generally protect themselves from free iron pools (PubMedID: 30236787).

(7) The authors state that a previous study demonstrated “GSH at 3.24 μmol·g-1 Hb, GSSG at 1.54 μmol·g-1 Hb, GSSG/GSH ratio at 0.56%)”. Based on the units and values this ratio cannot be real. In fact, a typical GSH/GSSG ratio is at least 200.

(8) The ratios calculated based on GSH and GSSG levels can be very misleading. They can be interpreted by several different ways, indicating oxidative or reductive stress or even unchanged redox state (PubMedID:30019441).

Intro: are there any other studies from a different sport on the topic? I think that it could be enriched. For instance, does endurance training affect exercise-induced acute changes and steady-state antioxidant enzyme activity (PubMedID: 28544643)?

Methods (2.5.3): were the samples for the GSH and GSSG measurement derivatized with N-ethylmaleimide? If not, this should be acknowledged as a major limitation of the study (PubMedID: 23928499).

Discussion: the authors state that “The decrease in GSH and the increase in GSSG during exercise can be explained by an increase in methaemoglobin formation”. If I am not wrong, you measured GSH in lymphocytes. If this is the case, revise as appropriate. If you used RBCs, then you should explain why units are in “protein” and not “Hb” (Table 2).

Author Response

Reviewer 4

The submitted work aimed to investigate the potential differences between professional and amateur cyclist in terms of endogenous antioxidant enzymes profile. Based on their findings, the authors argue that there is no well-defined endogenous antioxidant enzyme profile between the two levels of cyclists, as demonstrated by the higher CAT, GSSG and GSSG/GSH ratio and the lower SOD levels between PRO compared to AMA. With regard to performance, within the professional group, there was a reverse relationship between GSSG/GSH ratio levels and moderate and submaximal exercise performance. Please find next some comments on the paper:

Comment:  We thank the reviewer for their constructive and helpful feedback on our manuscript. We have replied to each specific comment in the section below and have introduced the corresponding edits into the manuscript using Word’s track changes. 

Title: (1) I think that an “s” is needed in “cyclist”; (2) the title refers to antioxidant enzymes profile, however, two enzymatic (CAT and SOD) and two non-enzymatic (GSH and GSSG as well as their ratio) parameters were measured. Probably, the title should be revised accordingly.

Response: Amended. As suggested, we have modified the title of the article.

Abstract: (1) what does the word “parallel” stand for? (2) use the abbreviations PRO and AMA in their first mention.

Response: Amended. We have changed parallel by comparative and introduced the abbreviations PRO and AMA. Lines 23-24.

General comment: the authors argue that a main aim of the study was to investigate “…if differences found in these parameters [antioxidant enzymes] could explain differences in performance”. Taking into account that only some correlation analyses were performed and that the theoretical background for such a cause-and-effect relationship is lacking in the introduction, I think that it is vague to argue that a real relationship can be established in the context of the present study. Certainly, differences in performance between these two groups can definitely be explained by other factors apart from redox state.

Response: Thank you for your appreciation, but our intention in the introduction was to show the reader that there were differences between PRO and AMA (training), and that the imbalance between the production of free radicals produced by exercise and the neutralization of these by the endogenous antioxidant system can affect macromolecules (proteins, plasma membranes, etc.), which influence muscle fatigue. It is already known that the main differences between PRO and AMA, in terms of performance, come from the capacity for oxygen consumption and power production at different intensities, but we wanted to know whether the endogenous antioxidant system supported these factors.

There are several issues in the introduction as regards to the information about (exercise-induced) redox processes:

  • The references provided are sometimes either not suitable or outdated. For the health- and redox-related benefits of chronic exercise check PubMedIDs: 32174127 and 32192916.

Response: Following your suggestion, we have changed reference 4 to PubMedIDs 32192916.

  • The increased oxygen uptake during exercise is currently regarded not the main reason for ROS production during exercise (and thus not mitochondria, but NADPH oxidases) (PubMedID: 23915064).

Response: Following your suggestion, we have introduced a sentence that explains the concept of the main source of ROS, and we have introduced the PubMedID reference: 23915064. Lines 65-66.

(3) 2% of the oxygen consumed is not converted to ROS. The real percentage is much lower, approximately equal to 0,13% to 0,15% (PubMedID: 12237311).

Response: Following your suggestion, we have changed the percentage of consumed oxygen that is converted to ROS to 0.15% and introduced a new reference 9 PubMedID: 12237311. Line 63.

  • Based on kinetic rates, peroxiredoxins are currently regarded the most efficient antioxidant enzymes against hydrogen peroxide and they also serve for signaling purposes (PubMedID: 20919930). This family of antioxidant enzymes are neglected in the present study.

Response: As suggested, peroxiredoxins are very important for H2O2 neutralisation, but we had a small budget for this study, which did not allow us to select antioxidant markers that had been measured previously in other studies and compare them with our data. Due to budget constraints, we were unable to measure other parameters.

  • Reactive species are in the present manuscript widely regarded as detrimental molecules. Even hydrogen peroxide is mentioned as “…also harmful to cells…”. However, we now know that this is probably the most well-known reactive species, along with nitric oxide, featuring signaling properties (PubMedID: 24634836).

Response: As suggested, we have taken into account that some ROS have intracellular signaling functions, transcription factors and genes, but this manuscript focuses on the negative effects of ROS production on performance. In addition, nitric oxide acts as a vasodilator; therefore, we believe that ROS are necessary. But an imbalance between production and elimination mechanisms can lead to impaired performance. In addition, this deterioration may vary according to the competitive level of the athletes (AMA vs PRO).

  • As the authors correctly mention, Fenton reaction requires free iron. However, this is probably not the case during exercise and in health in general, taking into account that biological systems generally protect themselves from free iron pools (PubMedID: 30236787).

Response: As suggested, we have introduced a sentence to clarify that the end product of the Fenton reaction is not clear. We have included the PubMedID reference: 30236787. Lines 61.

(7) The authors state that a previous study demonstrated “GSH at 3.24 μmol·g-1 Hb, GSSG at 1.54 μmol·g-1 Hb, GSSG/GSH ratio at 0.56%)”. Based on the units and values this ratio cannot be real. In fact, a typical GSH/GSSG ratio is at least 200.

Response: Based on the comment made about GSSG/GSH (0.56%), we have removed this ratio, as it is not consistent with the GSH and GSSG data. In addition, I have checked the referenced article and it does not indicate how this ratio is calculated.

(8) The ratios calculated based on GSH and GSSG levels can be very misleading. They can be interpreted by several different ways, indicating oxidative or reductive stress or even unchanged redox state (PubMedID:30019441).

Response: The reference you have indicated is from a study that used pathological models with tumour cells, and we believe that it is not comparable with the sample of our study in healthy cells.  Another problem with the suggested study is that the groups of subjects studied were not balanced, so the results lose power. This same study indicates that GSH/GSSG ratio is an indicator of oxidative stress: "The GSH:GSSG ratio may be used as a marker of oxidative stress".

Intro: are there any other studies from a different sport on the topic? I think that it could be enriched. For instance, does endurance training affect exercise-induced acute changes and steady-state antioxidant enzyme activity (PubMedID: 28544643)?

Response: As suggested, we have included the effects produced by acute and chronic exercise on the activity of antioxidant enzymes. Lines 88-92 and 94-98.

Methods (2.5.3): were the samples for the GSH and GSSG measurement derivatized with N-ethylmaleimide? If not, this should be acknowledged as a major limitation of the study (PubMedID: 23928499).

Response: Yes, ethylmaleimide was used.

Discussion: the authors state that “The decrease in GSH and the increase in GSSG during exercise can be explained by an increase in methaemoglobin formation”. If I am not wrong, you measured GSH in lymphocytes. If this is the case, revise as appropriate. If you used RBCs, then you should explain why units are in “protein” and not “Hb” (Table 2).

Response: As indicated, our GSH measurements were performed on lymphocytes, therefore, we have removed the sentence related to methaemoglobin formation. Thank you for your input. We have introduced a new paragraph. Lines 328-339.

Author comment: We appreciate all the comments made on our manuscript, which helped improve it’s quality.

Round 2

Reviewer 1 Report

manuscript approved for publication

Author Response

Thank you very much for your response, and for the suggestions you have made on our manuscript that have made it possible to improve it.

Reviewer 4 Report

I commend the authors for their rapid response 
No further comments!

Great work!

Author Response

(The authors gave the same response as above.)
